# The G-Protein-Coupled Estrogen Receptor Agonist G-1 Mediates Antitumor Effects by Activating Apoptosis Pathways and Regulating Migration and Invasion in Cervical Cancer Cells

**DOI:** 10.3390/cancers16193292

**Published:** 2024-09-27

**Authors:** Abigail Gaxiola-Rubio, Luis Felipe Jave-Suárez, Christian David Hernández-Silva, Adrián Ramírez-de-Arellano, Julio César Villegas-Pineda, Marisa de Jesús Lizárraga-Ledesma, Moisés Ramos-Solano, Carlos Daniel Diaz-Palomera, Ana Laura Pereira-Suárez

**Affiliations:** 1Instituto de Investigación en Ciencias Biomédicas, Centro Universitario de Ciencias de la Salud, Universidad de Guadalajara, Guadalajara 44340, Mexico; abigail.gaxiola2327@alumnos.udg.mx (A.G.-R.); adrian.ramirez@academicos.udg.mx (A.R.-d.-A.);; 2División de Inmunología, Centro de Investigación Biomédica de Occidente (CIBO), Instituto Mexicano del Seguro Social (IMSS), Guadalajara 44340, Mexico; 3Departamento de Microbiología y Patología, Centro Universitario de Ciencias de la Salud, Universidad de Guadalajara, Guadalajara 44340, Mexico; christian.hernandez@academicos.udg.mx (C.D.H.-S.); julio.villegas@academicos.udg.mx (J.C.V.-P.); marisa.lizarraga@alumnos.udg.mx (M.d.J.L.-L.); moises.ramos@academicos.udg.mx (M.R.-S.)

**Keywords:** cervical cancer, GPER, G-1, RNAseq, invasion, migration

## Abstract

**Simple Summary:**

The role of the GPER in cancer is controversial due to its dual anti and protumor effects. Elevated GPER expression in cervical cancer has been associated with improved survival outcomes. In cervical cancer cell lines, the selective GPER agonist G-1 induces cell cycle arrest and apoptosis, although the precise molecular mechanisms behind these effects are not fully understood. This study explores the impact of GPER activation by G-1 on the transcriptome, cell migration, and invasion in SiHa cells, as well as in non-tumorigenic keratinocytes transduced with HPV16 E6 or E7 oncogenes. G-1 has been shown to exert antitumor effects by activating apoptotic pathways and modulating migration and invasion processes. The findings support the GPER as a promising prognostic marker and suggest that G-1 could be a valuable therapeutic tool for treating cervical cancer.

**Abstract:**

Background/Objectives: Estrogens and HPV are necessary for cervical cancer (CC) development. The levels of the G protein-coupled estrogen receptor (GPER) increase as CC progresses, and HPV oncoproteins promote GPER expression. The role of this receptor is controversial due to its anti- and pro-tumor effects. This study aimed to determine the effect of GPER activation, using its agonist G-1, on the transcriptome, cell migration, and invasion in SiHa cells and non-tumorigenic keratinocytes transduced with the HPV16 E6 or E7 oncogenes. Methods: Transcriptome analysis was performed to identify G-1-enriched pathways in SiHa cells. We evaluated cell migration, invasion, and the expression of associated proteins in SiHa, HaCaT-16E6, and HaCaT-16E7 cells using various assays. Results: Transcriptome analysis revealed pathways associated with proliferation/apoptosis (TNF-α signaling, UV radiation response, mitotic spindle formation, G2/M cell cycle, UPR, and IL-6/JAK/STAT), cellular metabolism (oxidative phosphorylation), and cell migration (angiogenesis, EMT, and TGF-α signaling) in SiHa cells. Key differentially expressed genes included PTGS2 (pro/antitumor), FOSL1, TNFRSF9, IL1B, DIO2, and PHLDA1 (antitumor), along with under-expressed genes with pro-tumor effects that may inhibit proliferation. Additionally, DKK1 overexpression suggested inhibition of cell migration. G-1 increased vimentin expression in SiHa cells and reduced it in HaCaT-16E6 and HaCaT-16E7 cells. However, G-1 did not affect α-SMA expression or cell migration in any of the cell lines but increased invasion in HaCaT-16E7 cells. Conclusions: GPER is a promising prognostic marker due to its ability to activate apoptosis and inhibit proliferation without promoting migration/invasion in CC cells. G-1 could potentially be a tool in the treatment of this neoplasia.

## 1. Introduction

Cervical cancer (CC) stands as one of the most widespread malignancies globally. Each year, over 660,000 women are diagnosed with cervical cancer, which causes more than 348,000 deaths [1]. Its primary causative factor lies in persistent infection with high-risk human papillomavirus types [2]. However, additional factors contributing to carcinogenesis, including estrogens and their receptors [3], also play crucial roles. The interaction of these elements regulates viral oncogene expression [4], the tumor microenvironment [5], cell proliferation, and metabolic changes [6,7], all of which contribute to CC development and progression.

Epidemiological evidence suggests that female hormones, such as estrogen, play a role in the development of HPV-associated cervical cancer [8]. Consistently, estrogen has been shown to be required for cervical cancer development in transgenic mice expressing HPV E6 and E7 oncoproteins. When estrogen levels are reduced, disease progression slows, and partial regression occurs [9]. The presence of estrogens within the cervical microenvironment has been associated with the upregulation of HPV-encoded viral oncoproteins E6 and E7. This upregulation subsequently leads to an increase in the expression of estrogen receptors [4]. Estrogens exert their effects through specific receptors, primarily nuclear estrogen receptors α and β (ERα and ERβ), as well as the G protein-coupled estrogen receptor (GPER) [10]. Additionally, the estrogen receptor α (ERα) plays a crucial role in cellular transformation and oncogenesis in CC [11].

The GPER, also known as GPR30, has emerged as a critical player in estrogen signaling, exerting its actions through rapid non-genomic pathways and traditional genomic mechanisms [12,13]. The GPER exhibits diverse cellular localization patterns, adding complexity to its functional roles in physiological and pathological contexts. Traditionally recognized as a membrane-bound receptor, the GPER at the plasma membrane functions as a classical G protein-coupled receptor (GPCR), mediating rapid, non-genomic signaling cascades upon ligand binding. These cascades play a crucial role in initiating downstream signaling events that regulate cellular processes, such as proliferation, migration, and survival [14]. Additionally, the GPER has been identified within the endoplasmic reticulum, where it may modulate calcium homeostasis and ER stress responses [15,16], thereby influencing cellular functions beyond the canonical GPCR signaling pathways. Furthermore, evidence supports the GPER’s nuclear localization, where it may function as a transcriptional regulator by directly modulating gene expression. This contributes to the regulation of diverse biological processes, including cell cycle progression and differentiation [13].

Elevated levels of the GPER have been observed in cervical cancer (CC), and these levels increase as the disease progresses [17]. Exceptionally high GPER expression in the early stages of the disease is linked to positive clinical outcomes and improved survival rates [18]. Moreover, a synthetic selective agonist called G-1, which targets the GPER in cancer-derived cell lines expressing the GPER, inhibits proliferation and mitochondrial membrane potential. This leads to cell cycle arrest and apoptosis [17,19]. However, there is no evidence of the GPER’s involvement in inducing migration and invasion events. The mechanisms underlying GPER-mediated effects in CC remain incompletely understood, with evidence suggesting both antitumoral and protumoral roles, depending on various contextual factors [20].

The intricate involvement of the GPER in CC pathogenesis underscores its potential as a novel therapeutic target for the management of this malignancy. A deeper understanding of the complex interplay between estrogen signaling, viral factors, and host cellular responses holds promise for developing targeted therapies aimed at disrupting key molecular pathways driving CC progression.

Hence, this study aims to elucidate the impact of GPER activation via its specific agonist G-1 on the transcriptomic profile and functional behaviors, including cell migration and invasion, in SiHa cells and non-tumorigenic keratinocytes transduced with the HPV16 E6 or E7 oncogenes. The goal is to understand the molecular mechanisms underlying GPER-mediated effects in CC.

## 2. Materials and Methods

### 2.1. Cell Culture

The cervical cancer cell line (SiHa) was obtained from the American Type Culture Collection (ATCC, Manassas, VA, USA), and non-tumorigenic human keratinocytes (HaCaT) cell lines transduced with E6 or E7 oncogenes, with pLVX as a control, were used. Cells were cultured in Dulbecco’s Modified Eagle Medium (DMEM GlutaMAX, Cat. No. 10566016, Gibco, Miami, FL, USA) supplemented with 10% FBS (Cat. No. 26140-079, Gibco), penicillin (10,000 U/mL), streptomycin (10,000 μg/mL), and amphotericin B (25 µg/mL) (Cat. No. 15240062, Gibco). Cells were incubated in a water-jacketed incubator at 37 °C under an atmosphere of 95% air and 5% CO_2_ in the culture medium until they reached 80% confluence.

### 2.2. HaCaT Cells Transduced with E6 or E7 from HPV-16

HaCaT cell lines transduced with E6 and E7 oncogenes, as well as the pLVX control, underwent cloning via endpoint PCR to amplify the E6 and E7 open reading frames (ORFs) from genomic DNA samples derived from biopsies of patients infected with HPV-16. The following primer sets were employed: HPV 16-E6, forward: 5′ CAG ACA TTT TAT GCA CCA AA 3′, and reverse: 5′ CTC CAT GCA TGA TTA CAG C 3′; HPV 16-E7, forward: 5′ TAG AGA AAC CCA GCT GTA ATC A 3′, and reverse: 5′ AGG ATC AGC CAT GGT AGA TTA T. Each of the amplified products was individually cloned into a pGEM-T Easy cloning vector and then used to transform TOP10 Chemically Competent *E. coli* via heat shock. The transformed bacteria were selected, followed by restriction analysis with EcoRI. The oncoprotein genomes were sequenced and verified against the reference sequences in GenBank (HPV 16, accession no. K02718; https://www.ncbi.nlm.nih.gov/genbank/, accessed on 17 August 2023), revealing only a single substitution in HPV 16 E6, as described in sequence AF402678 (268T > G). Finally, the ORFs of the oncogenes were subcloned into a pLVX-Puro lentiviral expression vector.

Lenti-X 293T cells were used to produce infectious viral particles for HaCat cell infection via the Lenti-X Lentiviral Expression system. These cells were transfected independently with the pLVX-Puro empty vector, pLVX-HPV16E6, or pLVX-HPV16E7 vectors using the Lenti-X HTX Packaging system. Forty-eight hours after transfection, supernatants containing infectious viral particles were collected, and the presence of viral particles was confirmed using Lenti-X GoStix. HaCaT cells were individually infected with 100 μL of each viral supernatant, and the transduced cells were selected using 1 μg/mL puromycin. E6 and E7 expression levels in HaCaT cells were quantified via a reverse transcription-quantitative PCR (RT-qPCR) using the same primers employed for cloning. RNA extraction was carried out using an RNeasy Plus Mini kit following the manufacturer’s protocol, after which the RT-qPCR procedure was performed. To demonstrate the expression of the E6 and E7 oncogenes, we analyzed by RNA-seq data as shown in Appendix A.

### 2.3. Stimuli

G-1 (Cat. No. 10,008,933 Cayman Chemical, Ann Arbor, MI, USA) was prepared using dimethyl sulfoxide (DMSO, Cat. No. D8418-100ML, Sigma, Burlington, MA, USA) according to the manufacturer’s guidelines and added at a concentration of 1 μM. DMSO was used as a vehicle control.

### 2.4. RNA Extraction

Cells were stimulated with G-1 and incubated for 4 h. The total RNA was extracted from SiHa and HaCaT-16E7 cells using the RNeasy Plus Mini kit (Cat. No. 74136, Qiagen, Inc., Venlo, The Netherlands), according to the manufacturer’s instructions. RNA was quantified by measuring the absorbance at 260/280 nm.

### 2.5. RNA-Seq

Next-generation RNA sequencing was performed with the Nova Seq 6000 Illumina platform managed by Novogene Bioinformatics Technology Co., Ltd. (Beijing, China). Samples were preserved and sent in RNAstable (Cat. No. 93221-001, Biomatrica, San Diego, CA, USA). The collected data were analyzed using Rstudio (Version 4.2.1) and Galaxy (Version 23.1) open-source platforms. The quality of the sequencing data (FASTQ files) was assessed using FASTQC (Version 0.12.1). Afterward, alignment to the *Homo sapiens* genome (Version 42) (GRCh38.p13) was performed using the Subjunc aligner from the Rsubread package (Version 3.19). BAM files underwent processing using featureCounts (Version 2.0.2). Read counts were normalized to FPKM (fragments per kilobase of exon per million mapped fragments) using the DESeq2 tool (Version 1.40.2).

### 2.6. Gene Set Enrichment Analysis

The data obtained from the DESeq2 tool were used to create two matrices. An initial matrix, generated in a tab-delimited text file, contained the normalized expression data in FPKM. Simultaneously, the second matrix, created as a cls file, contained phenotype identifications. Both matrices were loaded into the GSEA software (version 4.2.3). The “Hallmark” gene set database (h.all.v2023.2.Hs.symbols.gmt) was selected, along with the Chip Platform option (Human_Ensembl_Gene_ID_MSigDB.v2023.2.Hs.chip). Gene set permutation was chosen as the permutation type, and the “ratio of classes” parameter was selected for gene ranking configuration. To identify significantly enriched pathways, a threshold of FDR < 0.25 was used for pathway selection.

### 2.7. Immunofluorescence

To determine the expression of EMT-associated proteins, 20,000 cells (HaCaT-pLVX, HaCaT-16E6, HaCaT-16E7, and SiHa) were seeded onto 8-cell slides and cultured for 24 h. The medium was then replaced with a serum-free medium containing the G-1 stimulus, and the cells were incubated for an additional 24 h. Afterward, the cells were rinsed with PBS and fixed with 4% paraformaldehyde at room temperature for 10 min. Following fixation, the cells were permeabilized in PBS containing 0.2% Tween 20 (Cat. No. P1379-500ML) for 10 min at 37 °C. Subsequently, a blocking step was performed using PBS containing 10% FBS (Cat. No. 26140-079) and 1% BSA (Cat. No. A2153-50G) for 1 h at 37 °C. The antibodies employed were Vimentin (Cat. No. RV202; 1:500 dilution) and α-SMA (Cat. No. ab7817; 1:500 dilution). The primary antibody was incubated overnight, followed by a 2 h incubation with the anti-mouse (FITC, ab6785; 1:1000 dilution) secondary antibody. The nuclei were subsequently stained with DAPI (1:15,000) for 5 min, protected from light. Slides were examined using an Axio Imager 2 fluorescence microscope (Carl Zeiss, Göttingen, Germany), with specific filters, including Alexa Fluor (excitation: 495 nm, emission: 519 nm) and DAPI (excitation: 351 nm, emission: 461 nm). For the analysis, at least 5 different fields per sample were observed.

### 2.8. Migration and Invasion Assays

Migration and invasion assays were conducted using a 6-well Transwell system (Cat. No. 3428, Corning, NY, USA) with an 8 μm pore size filter. Cells (HaCaT-pLVX, HaCaT-16E6, HaCaT-16E7, and SiHa) were detached with trypsin 24 h after G-1 stimulation. For the migration assay, 50,000 cells were seeded in the upper chamber in 1 mL of serum-free medium, while 1.5 mL of medium supplemented with 10% FBS was added to the lower chamber as a chemoattractant. After 24 h of incubation at 37 °C, cells remaining on the upper surface of the membrane were removed with a cotton swab. The cells were then fixed with 4% formaldehyde (No. Cat. 47608, Sigma, USA) and stained with 4.5% methylene blue (No. Cat. 03978, Sigma, USA). Cells that migrated through the filter were visualized using an inverted microscope. For the analysis, a minimum of 5 fields were observed per sample.

The procedure for the invasion assay was similar, except that the Transwell inserts were pre-coated with 800 μL of matrigel (1:3 dilution, Cat. No. 356234, Corning, NY, USA) per well for 1 h before 100,000 cells were seeded onto the membrane.

### 2.9. Statistics

Statistical analysis was performed using the GraphPad Prism 8.0.2. All experiments were performed with at least two independent replicates, and the normal distribution of the data was assessed. Group differences were calculated using paired Student’s *t*-tests or Wilconx tests. A *p*-value < 0.05 was considered statistically significant.

## 3. Results

### 3.1. G-1 Induces Transcription of Genes and Enrichment of Pathways Associated with Proliferation, Apoptosis, Metabolism, and Metastasis in the SiHa Cell Line

Next-generation mRNA sequencing was conducted to explore the molecular mechanisms induced by GPER activation via its agonist, G-1, in the SiHa cervical cancer cell line. Differential gene expression analysis was performed using the selection criteria of −2 ≤ Log2 (fold change) ≥ 2 and a *p*-value < 0.05, revealing overexpressed and under-expressed genes (Figure 1a). To identify the pathways modulated by G-1, an enrichment analysis was performed using the Hallmark gene collection from the version 4.2.3 of the GSEA software, with an FDR q value < 0.25 used for confident selection of regulated pathways. The pathways modulated by G-1 involve multiple processes, including proliferation and apoptosis (TNFα signaling pathways via NFκB, response to UV radiation, mitotic spindle formation, cell cycle arrest in G2/M, response to misfolded proteins, protein secretion, and IL-6/JAK/STAT3). Additionally, processes related to metabolism (oxidative phosphorylation) and metastasis (angiogenesis, epithelial-mesenchymal transition, and TGFβ signaling) were also affected (Figure 1b).

To identify the genes involved in the molecular mechanisms related to the pathways mentioned above, the differentially expressed genes (DEGs) were ranked by fold change and displayed in a heat map. These genes were then classified based on their association with proliferation and cell migration functions. These genes were then further categorized based on their protumor, antitumor, protumor/antitumor, or unknown roles in cancer (Figure 1c). The gene with the highest expression, *PTGS2*, codes for the cyclooxygenase-2 protein (COX-2) which is classified as both protumor and antitumor. Furthermore, numerous genes are associated with cell proliferation. Notably, about half of these genes may inhibit cell proliferation, including *FOSL1*, *TNFRSF9*, *IL1B*, *DIO2*, and *PHLDA1*, as well as other downregulated genes with protumor effects.

On the other hand, most of the genes involved in cell migration exhibit tumor-promoting effects. Notably, DKK1 expression was increased, and this gene is known to inhibit the migratory capacity of cells. Interestingly, *IQCN* was found to be downregulated; however, no reports currently link this gene to cancer.

### 3.2. G-1 Increases Vimentin Expression without Altering α-SMA Levels

Since metastasis-associated pathways were identified in the SiHa cell line, we evaluated the effect of the G-1 selective GPER agonist on the migratory and invasive capacities of cervical cancer cells to corroborate the functional effects of GPER activation on these processes. The expression of mesenchymal phenotype-associated proteins, vimentin, and α-SMA was evaluated by immunofluorescence. Stimulation with G-1 increased vimentin protein expression but had no effect on α-SMA expression (Figure 2a). G-1 stimulation did not induce significant changes in migration and invasion capacity; however, it showed a slight tendency to reduce these abilities (Figure 2c,d).

### 3.3. G-1 Modulates Vimentin and α-SMA Expression as Well as Invasion Processes in Keratinocytes

Since the SiHa cell line expresses both E6 and E7 oncogenes, and previous work from our research group has shown that E6 or E7 individually promote different GPER expression patterns, we observed particularly higher nuclear expression of GPER in the presence of E7 [4]. This suggests that E7 may play a significant role in modulating GPER localization and potentially influencing genomic signaling, which could differ from the non-genomic signaling induced when the GPER is activated at the plasma membrane by its agonist, G-1. This underscores the importance of investigating the distinct roles of each oncogene in the modulation of GPER expression and its related pathways.

We evaluated the effect of the G-1 selective GPER agonist on the migratory and invasive capacities of keratinocytes (HaCaT-pLVX, HaCaT-16E6, and HaCaT-16E7). The expression of mesenchymal phenotype-associated proteins, vimentin, and α-SMA was evaluated by immunofluorescence. Stimulation with G-1 reduced the protein expression of vimentin in keratinocytes transduced with the E6 or E7 oncogenes in contrast to non-transfected cells (Figure 3a). Regarding α-SMA expression, G-1 did not induce significant differences in HaCaT keratinocytes expressing oncogenes. However, a decrease in α-SMA expression was observed in keratinocytes that were free of oncogenes (Figure 3b). Cell migration and invasion potential were evaluated through a transwell chamber assay to corroborate the functional effect of G-1. Stimulation with G-1 did not induce significant changes in migration capacity in any of the cells (Figure 3c). However, the invasion capacity showed significant and distinct differences between these cells. G-1 induced higher invasion in HaCaT-16E7 cells compared to HaCaT-16E6 cells (Figure 3d).

### 3.4. G-1 Triggers Gene Transcription and Activates Pathways Associated with Proliferation, Apoptosis, Metabolism, and Metastasis in the HaCaT-16E7 Cell Line

Since keratinocytes expressing the E7 oncogene exhibited increased migratory capacity when stimulated with G-1, we deemed it crucial to gain a more comprehensive understanding of the mechanisms regulated by this treatment in the context of E7 expression. Consequently, we performed next-generation mRNA sequencing to elucidate the molecular pathways associated with GPER activation via its agonist, G-1. To identify differential expression, we applied selection criteria of −2 ≤ Log2 ≥ 2 and a *p*-value < 0.05, resulting in the identification of both downregulated and upregulated genes (Figure 4a). Fewer genes with differential expression were observed compared to the SiHa cell line.

Using the same analysis strategy applied to the SiHa cells, we used this approach to analyze pathway enrichment in the HaCaT-E7 cell line. Similar to the SiHa cells, GPER activation in HaCaT-E7 cells led to the enrichment of TNFα signaling pathways via NFκB, the UV radiation response, mitotic spindle formation, and cell cycle arrest in G2/M. It also enriched pathways related to the response to misfolded proteins, oxidative phosphorylation, protein secretion, IL-6/JAK/STAT3 signaling, angiogenesis, epithelial-mesenchymal transition, and TGFβ signaling. Additionally, in HaCaT-E7 cells, G-1 enriched pathways were strongly associated with cell death mechanisms, including ROS generation, P53 activation, DNA repair, and apoptosis (Figure 4b).

We presented a heatmap illustrating the DEGs categorized by processes such as proliferation, cell migration, and metabolic functions (Figure 4c). These DEGs were further classified based on their cancer-related effects as protumor, antitumor, protumor/antitumor, or unknown (Figure 4c). In the HaCaT-16E7 cell line, genes with protumoral effects were predominantly observed. Four genes related to cell proliferation were classified, with *IGFBP3* classified as antitumor and *DUSP10* as protumor/antitumor. Conversely, most DEGs associated with migration exhibited protumoral effects, although *DKK1* and the low expression of *COL20A1* may negatively regulate this process. Additionally, genes involved in cell metabolism were identified, with *ABCA1* potentially having both protumoral and antitumoral functions. Notably, as in SiHa cells, *DKK1* also showed elevated expression, which could inhibit cell migration. Interestingly, the downregulated expression of *IQCN* was observed in both cell lines. We selected the genes IL6, EREG, DKK1, and PDK4, and validated their expression using qPCR as shown in Appendix A.

## 4. Discussion

The effects of estrogen exposure on hormone-dependent tumor progression are widely documented. However, their precise role in cancer is complex due to the numerous receptors and cofactors with which they may interact. The mechanism by which estrogen contributes to cancer is unclear. Estrogen acts as a direct carcinogen and, therefore, may contribute to the initiation of lesions [21]. Additionally, estrogen is a known mitogen that could promote tumor growth [22]. Estrogens can also affect the efficiency of viral gene expression in HPV infections [23]. Estrogen not only initiates but also sustains and promotes the malignant progression of cervical cancer, according to a study conducted on an HPV transgenic mouse model [9]. Therefore, HPV and estrogens are essential factors for the development of cervical cancer. Estradiol has been observed to increase the expression of the E6 and E7 oncogenes of HPV 16 and 18 in cervical uterine cancer cells. These oncogenes, when present in non-tumorigenic keratinocytes, promote a significant increase in GPER protein expression [4].

The role of the GPER-driven signaling pathway, one of the most recently described estrogen receptors, in neoplasms is controversial due to its activation having both anti and protumorigenic effects. Therefore, the involvement of the GPER in the development and progression of cancer is still a subject of research and debate. Another critical finding suggesting the strong involvement of the GPER in cervical cancer is its strong overexpression in tumor tissue biopsies, compared to premalignant lesions (CIN I-III) and non-tumorous cervical tissue [17]. Additionally, increased GPER expression correlates with p16 and p53, and this is associated with improved survival and recurrence-free survival in early-stage cervical cancer [18]. This suggests a potential role for the GPER as a tumor suppressor in this disease; however, more information about GPER signaling pathway activation is needed to fully understand its effects on tumor progression.

In this study, to elucidate the molecular mechanisms induced by selective GPER activation through G-1, we analyzed the transcriptome of the SiHa cell line derived from cervical squamous cell carcinoma, which accounts for approximately 80% of cervical cancer cases, and of a non-tumorigenic keratinocyte cell line transduced with the E7 oncogene of HPV 16 to simulate an early lesion of this neoplasm.

The pathways enriched by G-1 stimulation in both cell lines modulate cell proliferation, apoptosis, cell migration, and metabolism. It is important to note that in SiHa cells, we identified more differentially expressed genes that can promote or inhibit cell proliferation than in HaCaT-16E7 cells. This finding is consistent with other studies reporting that GPER activation inhibits proliferation in cervical cancer (CC) [17,19].

The gene with the highest expression in SiHa is PTGS2, which codes for the cyclooxygenase-2 protein (COX-2) involved in tumor progression. In cervical cancer, however, a close relationship exists between p53 and COX-2 activation in the induction of apoptosis, mediated by ERK1/2 [24]. Importantly, the mechanism by which GPER can inhibit cervical cancer cell proliferation in vitro involves sustained activation of the ERK1/2 pathway through EGFR [19]. Similarly, COX-2 expression and ERK1/2 activation in ovarian cancer are essential in p53-dependent apoptosis [25]. Another gene differentially expressed by G-1 in SiHa is FOSL1. It has been shown not only to repress cervical cancer cell proliferation but also to promote apoptosis and induce S-phase cell cycle arrest. Additionally, it contributes to inhibiting the Warburg effect through STAT1 upregulation of p53 signaling [26]. Similarly, the PHLDA1 gene is upregulated in SiHa, and p53 expression has been observed to lead to the apoptosis of HeLa cells [27]. Another DEG is IL6, which potentiates TNF-α or TRAIL-dependent apoptosis by upregulating death receptors DR-4 and DR-5 through p53 activation in various cancers [28].

It is well known that GPER activation induces the release of calcium ions, serving as ubiquitous second messengers in cellular communication. The subsequent rise in cytosolic Ca2+ concentration initiates a cascade of downstream signaling events, modulating various cellular processes [29]. Continuous abnormal elevation of intracellular calcium concentrations could suppress cell proliferation and the initiation of apoptosis [30]. In G-1-stimulated MCF-7 breast cancer cells, calcium released into the cytosol activates the misfolded protein response (UPR), which is predominantly cell-death oriented, involving the phosphorylation of pERK and the subsequent activation of the JNK kinase, directly inducing apoptosis [16]. In cervical cancer cells, endoplasmic reticulum stress increases the UPR response and leads to tumor cell death through simultaneous autophagy induction by activating the NF-κB pathway [31]. GPER activation by G-1 induces cell cycle arrest in the G2/M phase through the upregulation of cyclin B [19]. As mentioned above, this agrees with our results, as the response to misfolded proteins, the NF-KB pathway, and G2/M phase cell cycle arrest were G-1-enriched pathways in SiHa and HaCaT-16E7 cells. We observed G-1-induced overexpression of the DIO2 gene in SIHA cells. It has also been reported that in papillary thyroid carcinoma cells, the elevated expression of this gene decreases cell cycle proteins, favoring G2/M phase arrest and reducing proliferative capacity [32].

Conversely, it is widely recognized that oncogenic HPVs exert a significant influence on gene expression to perpetuate tumor progression. HPVs have been documented to inhibit the expression of the tumor suppressor gene DKK1, which hinders cisplatin-induced apoptosis, leading to the survival of HPV-positive cancer cells [33]. We found increased expression of this gene in G-1-stimulated HaCaT-16E7 and SiHa cell lines, suggesting that DKK1 reactivation could direct neoplastic cells toward cell death. It is relevant to note that in both cell lines, DKK1 exhibits elevated expression, which may inhibit cell migration. Interestingly, the downregulated expression of IQCN was observed in both cell lines; however, to date, no reports link this gene to cancer. Its function as a microtubule stabilizer, however, has been established [34]. All these results confirm that the GPER signaling pathway is essential for reducing cell proliferation and inducing apoptosis, providing evidence to consider the GPER as a tumor suppressor and G-1 as a potential therapeutic agent.

While we have provided evidence for an antitumor effect of the GPER, it has also been documented that estrogen/GPER signaling causes genomic instability by inducing double-strand breaks, which can lead to carcinogenesis in high-risk HPV-infected endocervical columnar cell lines [35]. We found that in HaCaT-16E7, G-1 induces pathways associated with DNA repair, p53 pathway activation, and apoptosis, suggesting that these DNA breaks may reflect the induction of cell death mediated by GPER-enriched pathways. Treatment with G-1 has shown the potential to induce apoptosis in certain cell lines via the p53 pathway. However, apoptosis induction through this pathway is ineffective in the HaCaT cell line, which harbors mutations in the TP53 gene. The mutations in p53 in HaCaT result in the loss of its normal tumor suppressor function, preventing the activation of p53-dependent apoptotic mechanisms [36]. In the HaCaT cell line, G-1 does not induce apoptosis because the TP53 gene is mutated, which blocks the p53-dependent apoptosis pathway. Therefore, in other cell lines with functional TP53, G-1 could induce apoptosis through p53 activation [17]. These results confirm that the ability of G-1 to induce apoptosis is closely linked to the functionality of the p53 pathway.

Another important point is that G-1 can act independently of the GPER by binding to tubulin at the colchicine binding site, which destabilizes microtubule dynamics during mitosis in endothelial cells [37], ovarian cancer [38], breast cancer [39], and T-lineage leukemia [40]. This action leads to the induction of apoptosis and the inhibition of tumor growth.

As mentioned, oncogenic HPVs regulate protumoral processes through their oncogenes, leading to the progression to metastasis. The E7 oncogene has been observed to stimulate cell invasion by increasing MMP-9 expression, resulting in extracellular matrix degradation and cell invasion [41]. Furthermore, in cervical adenocarcinoma, estrogen signaling through the GPER promotes claudin-1 production, contributing to cytoskeleton remodeling and enhancing malignant capabilities in cervical adenocarcinoma cells [42,43]. GPER knockdown increased the formation and number of filopodia in HeLa cells [44]. This suggests that GPER knockdown may promote features favoring tumor progression, such as cell migration and invasion, by increasing filopodia activity, cell membrane extensions involved in cell motility, and interaction with the extracellular environment. Importantly, our data show that G-1 enhances invasiveness only in E7-transduced keratinocytes, a phenomenon not observed in SiHa tumor cells. This observation could be explained by the fact that GPER expression levels are low in cervical intraepithelial neoplasia compared to cancer tissue [17], suggesting that the antitumor regulation mediated by the GPER may be less effective in these contexts.

Another way to assess the invasive capacity of tumor cells is by measuring vimentin expression. However, an increase in this intermediate filament has been reported in situations of oxidative stress and mitochondrial damage associated with apoptosis [45]. In cervical cancer, G-1 decreases mitochondrial membrane potential and increases apoptosis [17]. In line with this evidence, our results indicate that in SiHa cells, G-1 increases vimentin protein expression. However, the absence of changes in migrating and invading cell numbers reflects the induction of cellular stress and, consequently, apoptosis.

## 5. Conclusions

Our findings highlight that GPER activation by G-1 plays a vital role in tumor suppression and suggest G-1 as a potential therapeutic agent in cancer treatment. We also found that G-1 can affect cell invasiveness in E7-transduced cells, which could be related to GPER expression levels at different disease stages. Additionally, G-1 has been shown to directly affect microtubule dynamics, contributing to its ability to induce apoptosis and suppress tumor growth. These results suggest a broad therapeutic potential for G-1 in several types of cancer, independent of its interaction with the GPER. In summary, our findings provide new insights into the mechanisms of action of the GPER and the therapeutic potential of G-1 in cancer treatment. However, further investigation is needed to fully understand the magnitude of these effects and their therapeutic viability in cervical cancer.

## Figures and Tables

**Figure 1 cancers-16-03292-f001:**
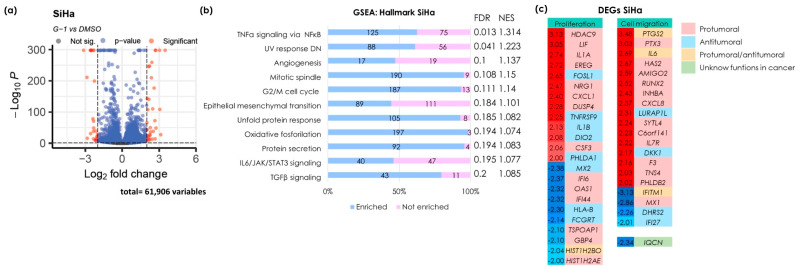
The identification of differentially expressed genes and enriched pathways modulated by G-1 in the SiHa cell line. (**a**) A volcano plot illustrating DEGs with fold changes of −2 or lower and 2 or higher, and a *p*-value of less than 0.05. Red circles on the right indicate upregulated genes, while red circles on the left indicate downregulated genes. (**b**) An enrichment analysis of the Broad Institute’s Molecular Signature Database Hallmark gene collection was evaluated using the version 4.2.3 of the GSEA software. The left panel shows statistically significant pathway names and the right panel shows the false discovery rate (FDR) and normalized enrichment score (NES) values. The numbers in the bar graph indicate the number of enriched (blue) and non-enriched (pink) genes within each pathway. An FDR < 0.25 was set as the selection criteria. (**c**) A heatmap of DEGs selected by −2 ≤ Log2 ≥ 2 and a *p*-value < 0.05. The numbers on the left, shown in red and blue, represent the fold change values. Red numbers indicate gene upregulation, while blue numbers signify gene downregulation. Color coding indicates the detailed analysis of previous publications related to each gene.

**Figure 2 cancers-16-03292-f002:**
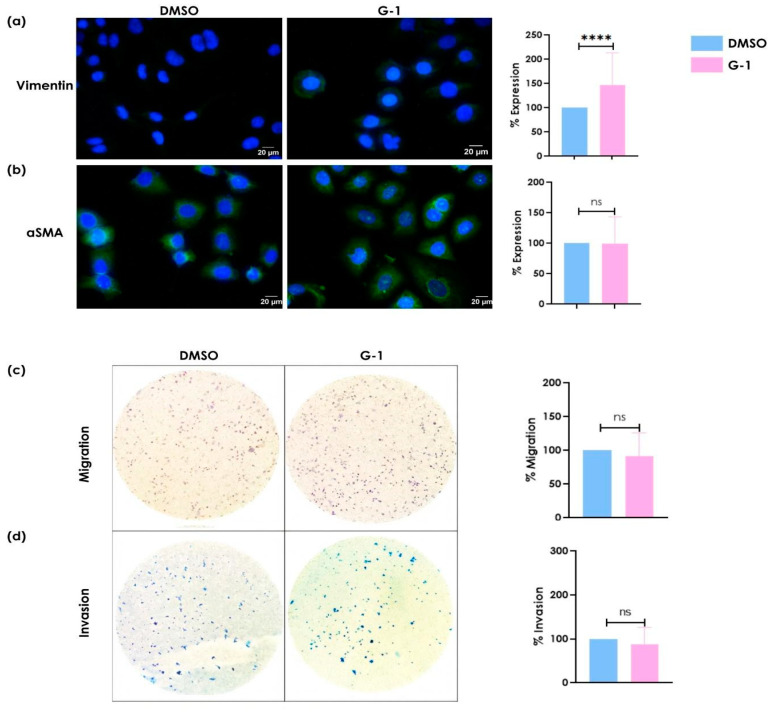
The effect of GPER activation on the expression of (**a**) vimentin and (**b**) αSMA and the induction of (**c**) migration and (**d**) invasion in SiHa cell line. This was cultured and stimulated with 1 μM of G-1 for 24 h. Immunofluorescence was performed using a secondary antibody conjugated to FITC (green) and DAPI staining (blue). Merged images are presented at 40×. The migration/invasion assays were performed in transwell chambers. The results are shown as the mean ± SD (**** *p* ≤ 0.0001; ns: not significant).

**Figure 3 cancers-16-03292-f003:**
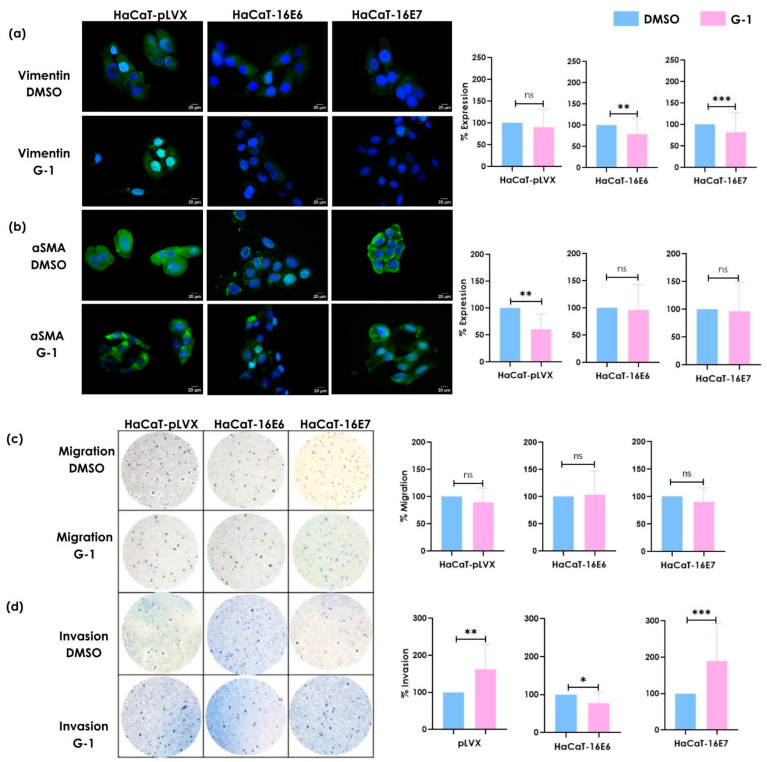
The effect of GPER activation on the expression of (**a**) vimentin and (**b**) αSMA and the induction of (**c**) migration and (**d**) invasion in HaCaT-pLVX, HaCaT-16E6, and HaCaT-16E7 cell lines. These were cultured and stimulated with 1μM of G-1 for 24 h. Immunofluorescence was performed using a secondary antibody conjugated to FITC (green) and DAPI staining (blue). Merged images are presented at 40×. The migration/invasion assays were performed in transwell chambers. The results are shown as the mean ± SD (* *p* ≤ 0.05; ** *p* ≤ 0.01; *** *p* ≤ 0.001; ns: not significant).

**Figure 4 cancers-16-03292-f004:**
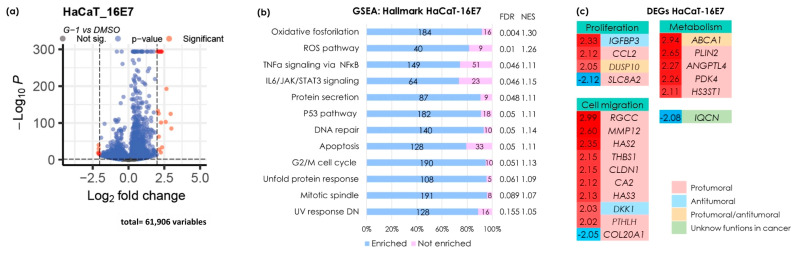
Identification of differentially expressed genes by G-1 in HaCaT-16E7. (**a**) A volcano plot illustrating DEGs with fold changes of −2 or lower and 2 or higher, and a *p*-value of less than 0.05. Red circles on the right indicate upregulated genes, while red circles on the left indicate downregulated genes. (**b**) An enrichment analysis of the Broad Institute’s Molecular Signature Database Hallmark gene collection was evaluated using the version 4.2.3 of the GSEA software. The left panel shows statistically significant pathway names and the right panel shows the false discovery rate (FDR) and normalized enrichment score (NES) values. The numbers in the bar graph indicate the number of enriched (blue) and non-enriched (pink) genes within each pathway. An FDR < 0.25 was set as the selection criteria. (**c**) A heatmap of DEGs selected by −2 ≤ Log2 ≥ 2 and a *p*-value < 0.05. The numbers on the left, shown in red and blue, represent the fold change values. Red numbers indicate gene upregulation, while blue numbers signify gene downregulation. Color coding indicates the detailed analysis of previous publications related to each gene.

## Data Availability

The datasets used and/or analyzed during the current study are available from the corresponding authors upon reasonable request.

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
