# Peer review of "The G-Protein-Coupled Estrogen Receptor Agonist G-1 Mediates Antitumor Effects by Activating Apoptosis Pathways and Regulating Migration and Invasion in Cervical Cancer Cells"

_cancers, 2024, doi:10.3390/cancers16193292_

Round 1

Reviewer 1 Report

Comments and Suggestions for Authors

Check line 92

Why Hacat and not human non immortalized E6/E7 trasduced keratinocytes were used?

Please, show E6 and E7 levels by WB

All the figures have a poor quality, expecially the IF figures. Please, replace with focused images.

Why authors used unpaired Student t-tests or Mann Whitney U tests instead paired Student t-tests or Wilconx tests, in particularly for the comparisons among the same cell line treated with  DMSO or G1. Please, redo the statatistical analyses.

The most up- and down-modulated DEGs should be confirmed by Real time PCR. Please, provide data about fig 1 and fig 4.

Please, provide y-labels  for fig 2 graphs and scale bars for IF images.

Why pLVX induced a-SMA G1 reduction and invasion % increase ? How you can distinguish the E7 effect  from the empty pLVX  effect on migration? Please, provide scale bars for fig 3a IF images and better images for fig 3c and 3d.

Comments on the Quality of English Language

Please, provide  a certificate for English approval.

Reviewer 2 Report

Comments and Suggestions for Authors

The aim of the research conducted by Abigail Gaxiola-Rubio and colleagues is to determine the effect of GPER activation by G-1 on the transcriptome, cell migration and invasion in SiHa cells, as well as in non-cancerous keratinocytes transduced with HPV16 E6 or E7 oncogenes. In my opinion, the experiments presented in the submitted manuscript are characterized by a high level of substantive content. The researchers correctly and reliably planned and performed a series of experiments aimed at determining the role of GPER in cervical cancer. The variety of research techniques used and the innovative approach to the presented topic deserve praise. The literature was selected correctly. The results of the studies are described carefully and clearly. The exceptions are figures 1b, c and 4b, c, the quality of which is too low and therefore illegible - please correct these figures. After reading the content of the manuscript, I have no major comments and recommend the presented manuscript for publication in the Cancers.

Round 2

Reviewer 1 Report

Comments and Suggestions for Authors

Authors provide sufficient informations and comments to consider the paper suitable for the publication